# Micro/Nanostructured Coating for Cotton Textiles That Repel Oil, Water, and Chemical Warfare Agents

**DOI:** 10.3390/polym12081826

**Published:** 2020-08-14

**Authors:** Jihyun Kwon, Hyunsook Jung, Heesoo Jung, Juno Lee

**Affiliations:** 4th R&D Institute-6th Directorate, Agency for Defense Development, Yuseong-Gu, Daejeon 34186, Korea; jhkwon8@add.re.kr (J.K.); junghs@add.re.kr (H.J.); hsjung@add.re.kr (H.J.)

**Keywords:** OmniBlock, chemical warfare agent, textile, coating

## Abstract

Using a lotus leaf as our model, we fabricated an extremely low surface energy micro/nanostructured coating for textiles that repel oil, water, and chemical warfare agents (CWAs) using a simple process that is suitable for large scale production. This coating, called “OmniBlock”, consisted of approximately 200-nm silica nanoparticles, tetraethylorthosilicate, 3-glycidoxypropyl trimethoxysilane, and a perfluorooctanoic acid-free fluoropolymer (Fluorolink S10) that was cross-linked between Si-O-Si groups via a sol-gel process. The perfluorooctanoic acid-free fluoropolymer-coated silica nanoparticles were simply applied to the surface of a cotton fabric by a dip–dry–cure process, forming dense, continuous, and uniform layers of OmniBlock coating. OmniBlock modified the surface of the cotton fibers, creating a rough, high surface area uniform coating with many micro-crevasses. As a result, n-dodecane, water, and CWAs beaded up without wetting the surface, exhibiting large contact angles of 154° for water and 121° for n-dodecane, with a small shedding angle of 5° and contact angle hysteresis of 3.2° for water. The designed coating showed excellent liquid repellence properties against three types of CWAs: 129°, 72°, and 87° for sulfur mustard (HD), soman (GD), and VX nerve agents, respectively. Furthermore, OmniBlock coating shows good mechanical properties under tensile strength and wash tests. This remarkable ability to repel CWAs is likely to have potential military applications in personal protective equipment systems requiring self-cleaning functions.

## 1. Introduction

Liquid repellent surfaces with self-cleaning properties are of current research interest, which have been mainly inspired by lotus leaf, known as the Lotus effect [1,2,3]. The effect occurs due to the micro/nanostructures on the leaf that are covered with hydrophobic waxes, providing a low surface free energy [4]. This wax combined with the micro/nanostructures results in a hierarchical double structure in which air pockets form on the surface, repelling water by dramatically increasing the contact angle of water droplets [2]. As a result, water droplets easily capture dust particles by rolling on the leaf, and in doing so, exhibit self-cleaning properties. Replicating these self-cleaning properties can be especially useful for chemical protective clothing, which can require less or no laundering, resulting in an extended wear life, shelf life, and an enhanced protective performance of the clothing. Learning from nature, researchers have made a variety of attempts: plasma etching [5,6], lithography [7,8,9], chemical vapor deposition [10], self-assembly [11,12], layer-by-layer deposition [13,14,15], polymerization reaction [16,17], and sol-gel reaction [18,19] in order to modify a surface’s resistance to wetting by high surface tension liquids like water and low surface tension liquids like an oil (omniphobicity). For instance, researchers have developed self-cleaning textile by combining low surface energy materials such as organic silanes [20,21], fluorinated silanes [22,23], alkyl amines [24], and creating surface roughness [25]. However, these methods typically require expensive, high-vacuum equipment, complicated fabrication processing, and usually the resultant surfaces are easily damaged [26,27,28]. Therefore, these types of approaches are not practical on an industrial scale and other strategies are required to achieve large-area omniphobic surfaces for mass fabrication.

In recent years, sol-gel chemistry, a solution process, is an easy and effective technique that has been adapted to large scale industrial applications because it can coat large-area surfaces at a low cost and with improved durability [29,30,31,32]. Accordingly, the sol-gel chemistry is often investigated for applying hydrophobic coating to fabrics. The main process involves hydrolysis with silica precursor at low temperatures and fast synthesis times. The reactions involve three steps: (1) hydrolysis of the precursor source in the acidic/basic conditions, (2) alcohol condensation, and (3) water condensation [30]. After the condensation process, particles are added to substrate surface. Hydrophobic sol-gel solutions can be deposited onto the surface via spray coating [31], dip coating [32] and thermal curing step [33,34]. Indeed, low-energy surfaces of fabric can be prepared using sol-gel reactions of repellent material such as silica nanoparticles, fluorinated alkylsilane, and perfluoroalkyl compounds [35,36,37]. Fabric coating with silica-based nanosols with perfluoroalkyl compounds was used to get functional properties that are super-hydrophobic and oleophobic. For instance, Ma et al. have successfully demonstrated a ‘non-sticky’ superhydrophobic surface using a sol-gel processed alumina gel film and a fluoroalkyl phosphonic acid treatment [38]. Wei et al. also developed omniphobic self-cleaning surfaces based on a fluorinated hybrid nanocomposite via a nanosilica sol and a fluorine-containing compound [39]. Satoh et al. reported water repellent textile coating with relatively low fluorine contents using inorganic-organic composed perfluroalkylsilane by sol-gel process [40]. Recently, commercialized sol-gel coatings contain micro/nanoparticles in their formulation that can provide an additional layer of surface roughness to repel low surface energy liquids [41].

Repelling oil, water, and chemical warfare agents (CWAs) from the surfaces of textiles is a necessity for safely protecting soldiers in a battlefield. For the past 40 years, textiles used for chemical protective clothing have been coated with polytetrafluoroethylene (PTFE) to minimize surface wetting of the fabric. However, PTFE was phased out in 2015 because its manufacturing requires perfluorooctanoic acid (PFOA) also known as C8, which has been linked to several health concerns, including cancer, and persists in the environment and living organisms [42,43,44]. Therefore, there have been a growing number of PFOA-free coating technologies [42,45,46,47]. Furthermore, low surface tension liquids such as oils and CWAs can easily wet C8-treated clothing after repeated wearing and washing, gravely concerning the soldiers who use it. The typical low surface tension value for n-dodecane is 25.3 mN/m, for mustard gas (HD) is 42.5 mN/m, for GD is 24.5 mN/m, and for VX is 31.3 mN/m [48,49]. For potential use in CWA’s protective fabric, coating requires materials to effectively remove both non-polar and polar chemicals. In particular, CWAs are polar, so that it is challenging to deal with omniphobic surfaces that repel both non-polar and polar liquids [50].

In this work, we fabricated a material called OmniBlock which displayed excellent liquid repellence properties against oil, water, and CWAs (HD, GD, and VX). We used tetraethyl orthosilicate (TEOS) to prepare approximately 200-nm silica nanoparticles (Si NPs), and then 3-glycidoxypropyltrimethoxysilane (GLYMO) and perfluorooctanoic acid (PFOA)-free fluoropolymer (Fluorolink S10) were cross-linked between Si-O-Si groups via a simple sol-gel chemistry to obtain PFOA-free fluoropolymer-coated Si NPs (Figure 1). In particular, Fluorolink S10 has two silane end groups and a relatively high fluorocarbon in comparison with common fluoroalkyl compounds, featuring better omniphobic coating and mechanical resistance [51]. By coating cotton fabric in a PFOA-free fluoropolymer-coated Si NPs solution via a dip–dry–cure process, a rough, high surface area oleophobic structure developed on the surface of the cotton fabric. The OmniBlock-coated cotton fabric displayed excellent liquid repellence properties to n-dodecane, water, and CWAs (HD, GD, and VX) with good mechanical properties. This work provides a new route for the large-scale production of liquid-repellent textile surfaces for chemical protective clothing and fabric industries.

## 2. Materials and Methods

### 2.1. Materials

Raw cotton fabric (plain weaved, 30 Ne, 126.3 g/m^2^) was purchased from a local fabric store. Tetraethyl orthosilicate (TEOS, 98%), isopropyl alcohol (IPA, 99.5%), 3-glycidoxypropyltrimethoxysilane (GLYMO, 98%), 1-methyl imidazole (1-MI, 99%) and ammonium hydroxide solution (NH_4_OH, 28%) were purchased from Sigma-Aldrich (St. Louis, MO, USA) and used without further purification. Fluorolink S10, n-dodecane, and ethanol were purchased from Solvay, Kanto Chemical (Tokyo, Japan), and Samchun Chemical (Seoul, Korea), respectively.

HD (2, 2′-dichloroethyl surfide), GD (3, 3-dimethyl-2-butyl methylphosphonofluoridate), and VX (O-ethyl-S-2(diisopropyl amino)ethyl methylphosphonothioate) (≥99%) were provided by the Chemical Analysis Test and Research Laboratory at the Agency for Defense Development (ADD, Daejeon, Korea). Due to their extreme danger and toxicity, extreme care was taken when handling CWAs. CWAs should only be handled by trained personnel using sufficient protective equipment and relevant safety procedures, as was done in this study.

### 2.2. Preparation of Si NPs Solution

Si NPs were synthesized according to the Stöber method as reported in the literature [52]. First, 10 mL of TEOS was dissolved in 50 mL of ethanol. Then, 12 mL of NH_4_OH (28%) in 50 mL of ethanol was added into the TEOS solution. After stirring for 24 h at room temperature, the prepared Si NPs were repeatedly rinsed with ethanol:H_2_O (1:1 *v*/*v*) and with ethanol to remove excess NH_4_OH. After washing, the resulting Si NPs (~200 nm) solution was dispersed in 100 mL of ethanol by ultrasonication (Power sonic 50, Hwashin Tech, Gwangju, Korea) for 30 min.

### 2.3. PFOA-Free Fluoropolymer-Coated Si NPs (OmniBlock)

PFOA-free fluoropolymer-coated silica nanoparticles, or OmniBlock, was prepared via the acid catalyst-based co-condensation [53,54,55] of Fluorolink S10, GLYMO, and TEOS with Si NPs (~200 nm). 1 mL of TEOS and 1 mL of GLYMO were dissolved in 5 mL of IPA. The pre-dispersed Si NPs solution (15 mL), 0.1 mL of Fluorolink S10, 0.58 mL of deionized water, and 1 mL of 1M HCl were added to the IPA mixture. The mixture was then stirred for 6 h at room temperature to obtain PFOA-free fluoropolymer-coated Si NPs. Finally, 1 mL of 1-methyl imidazole solution in IPA (10%) was added while stirring for an additional 10 min.

### 2.4. Dip–Dry–Cure OmniBlock Coating

A dip–dry–cure process was used to fabricate OmniBlock-coated cotton fabric. Cotton fabric (20 × 20 mm) was immersed into the PFOA-free fluoropolymer-coated Si NPs solution for 10 min. The cotton fabric was then dried for 10 min at 80 °C and finally cured for 60 min at 120 °C. Prior to coating, activation of the cotton surface was performed with an Ar plasma generated at 100 W RF power, 20 sccm (standard cubic centimeter) and 200 mTorr for 100 s. The overall fabrication process of OmniBlock-coated cotton fabric is shown in Figure 1.

### 2.5. Characterization of Surfaces

Scanning electron microscopy (SEM) was performed using a JSM-IT500HR instrument (JEOL, Tokyo, Japan) in high-vacuum mode operated at an accelerating voltage of 5 kV and a probe current of 35 A. X-ray photoelectron spectroscopy (XPS) measurements were performed with an Al Kα X-ray source using a K-Alpha+ spectrometer (ThermoFisher scientific, Waltham, MA, USA). Attenuated total reflection-Fourier transform infrared spectroscopy (ATR-FTIR) was performed using a Nicolet iS50 FT-IR (ThermoFisher scientific, Waltham, MA, USA) with the range from 3800–400 cm^−1^. Thermogravimetric analysis (TGA) was performed using a TGA Q500 apparatus (TA Instruments, New Castle, DE, USA) under a nitrogen atmosphere at a flow rate of 100 cm^3^/min. Additionally, TGA was performed in the air. The temperature range was from 25 to 800 °C with a heating rate of 10 °C/min.

### 2.6. Liquid Repellency Measurements

Contact angle measurements were carried out using a drop shape analyzer (DSA 100 instrument from KRŰSS, Hamburg, Germany) at room temperature. A 5-μL droplet for water (deionized) and n-dodecane and 3-μL droplet for CWAs (HD, GD, and VX) were used. We reported the average value of surface contact angles for at least five measurements after resting for 30 s, and the measurements were made on different areas of the coated sample. The Zimmermann method [56] was used to measure the shedding angles, applying a device consisting of a USB microscope (UM12, Vitiny USA) with an angle-adjustable manual stage (G1-80R87, DPIN). The distance between the syringe tip and fabric was fixed at 1 cm. Water and n-dodecane droplets (13 μL, each) were dropped onto the inclined surface, and then the minimum angles were measured at which the droplets of water and n-dodecane rolled off, respectively. For the contact angle hysteresis analysis, the advancing and receding angles were measured using the sessile method [57]. While a water droplet of 5.0 μL was suspended to the needle on the surface of OmniBlock-coated cotton fabric, additional water was spurred out to 7.5 μL and the advancing contact angle was measured. The receding angle was measured with additional suctions to 2.5 μL. The hysteresis was calculated from the difference between the advancing and receding angle.

### 2.7. Mechanical Properties

Universal testing machine Instron-5967 (Instron, Norwood, MA, USA) was used to examine tensile strength in accordance with the KSK0521 strip method. The sample size was 25 × 170 mm rectangular. The gap between the clamps was 75 mm and tension speed was 50 mm/min. For wash test, the samples (30 × 30 mm) were agitated by a magnetic stirrer at 300 rpm for 30 min in the 1% of Persil^®^ (Henkel, Dusseldorf, Germany) detergent solution, then were rinsed at 300 rpm for 10 min in the water (deionized) and dried at 60 °C for 10 min.

## 3. Results and Discussion

### 3.1. OmniBlock-Coated Cotton Fabric

The surface morphologies of raw cotton, Ar plasma-treated cotton, and OmniBlock-coated cotton fabric are shown in Figure 2. The raw cotton fabric in Figure 2a exhibits long, narrow surface striations somewhat aligned parallel to the length of the fiber. However, after activation by the Ar plasma treatment in Figure 2b, the surface of the cotton fabric was etched due to the ion bombardment from the Ar plasma [58,59,60], which enhanced the adhesion of the coating by increasing the contact area of the fiber [61]. Images of the OmniBlock-coated cotton fabric in Figure 2c indicate that the PFOA-free fluoropolymer-coated Si NPs appeared to form uniformly packed (Appendix A), continuous layers of coating, making the surface appear rougher and likely with a higher surface area. The size of Si NPs is approximately 200 nm (Figure 2d). The roughness of the OmniBlock-coated cotton fabric was significantly greater than of the raw cotton fabric (Appendix A).

XPS was used to qualitatively characterize the changes in the surface chemical composition of the cotton fabric before and after the OmniBlock layer coating was applied. For the raw cotton fabric, only two peaks corresponding to C and O elements were observed at 290 eV (C 1s) and 536 eV (O 1s), respectively, as shown in Figure 3. After coating with OmniBlock, three new distinctive peaks appeared at 688, 154, and 103 eV, which were attributed to the F 1s, Si 2s, and Si 2p signals, respectively [62,63,64]. This suggests that PFOA-free fluoropolymer-coated Si NPs have covered the surface of the cotton fibers. Additionally, ATR-FTIR spectra of OmniBlock-coated cotton fabric support the covalent attachment of the PFOA-free fluoropolymer to the coated Si NPs (Figure 2b). The peak appearing at 1100 cm^−1^ is associated with the asymmetric stretching vibration of Si-O-Si bonds [65,66]. Other peaks at 800 and 480 cm^−1^ are associated with a Si-O-Si bond for the bending and vibration, respectively [67]. The peak at 1145 cm^−1^ represents a Si-O-C bond [68], confirming the Si NPs attachment to the cotton surface. Reportedly, the degree of condensation of the silsesquioxane networks is high even when precursors bearing long alkyl chains have been used [69,70]. Thus, we believe that the sol-gel reaction between Si NPs and fluorinated alkoxysilane may lead to a high degree of condensation.

TGA was used to evaluate the weight loss of raw and coated cotton fabrics after heating to 800 °C at a rate of 10 °C/min, as shown in Figure 4. TGA and DTG (derivative thermogravimerty) data of 5% (T_5%_) and 10% (T_10%_) mass loss temperatures, the maximum-rate degradation temperatures (T_max1_, T_max2_), the residual percentages at T_max1_, T_max2_, and final residual percentages at 800 °C are listed in Appendix A. Under air atmosphere, the T_5%_, T_10%_ of raw cotton were 90 °C, 260 °C, and the final residue at 800 °C was 1.3%. As for OmniBlock-coated cotton, the T_5%_, T_10%_, and the final residue at 800 °C increased to 192 °C, 273 °C, and 9.6%. Additionally, under nitrogen atmosphere, the T_5%_ and T_10%_ of raw cotton were 75 °C, 273 °C, and the final residue at 800 °C was 16.3%. As for OmniBlock-coated cotton, the T_5%_, T_10%_, and final residue at 800 °C increased to 219 °C, 287 °C, and 23.9%, mainly due to the production of inorganic SiO_2_ from the degradation of OmniBlock, indicating that thermal stability was also improved by the OmniBlock coating in comparison with the raw cotton [71,72].

### 3.2. Liquid-Repellence Properties of OmniBlock-Coated Cotton Fabric

Excellent liquid-repellence properties on any surface are characterized by large contact angles (*θ* > 150°) and small shedding angles (*θ* < 10°) [73]. In this study, contact angles were measured in a static mode whereas shedding angles were dynamic since the angle was determined by the sliding of the water droplet on a continuously inclined surface [74]. As presented in Figure 5, water and n-dodecane produced spherical and floating droplets on the OmniBlock-coated cotton fabric. The static contact angle of 5 μL of water was measured at 154°. Besides water repellency, the coated cotton surface exhibited excellent repellency for the non-polar liquid (oleophobic), n-dodecane and for CWAs like HD, GD, and VX. It is noted that CWAs are polar but possess relatively low surface tensions [48]. The surface contact angle values with n-dodecane, HD, GD, and VX were 121°, 129°, 72°, and 87°, respectively. It should be noted that the Ar plasma-treated cotton fabric was fully wet with dyed water and n-dodecane, while the droplets of dyed water, n-dodecane and CWAs (HD, GD) floated on the OmniBlock-coated cotton fabrics (Appendix A). As depicted in Figure 6a, the shedding angle of water (13 μL) was 5° and of n-dodecane (13 μL) was 45° (Appendix A). Additionally, the advancing and receding angles were measured to obtain the contact angle hysteresis (Figure 6b) The contact angle hysteresis was 3.2°, obtained from the difference between the advancing angle and receding angle of water (advancing angle of 151.5° and receding angle of 148.3°), confirming that OmniBlock-coated cotton fabric is superhydrophobic. Robin M. Bär et al. report that robust superhydrophobic surface with various coating methods. The maximum value of 160.2° shows superhydrophocity, but the contact angle hysteresis is measured above 10° [73].

To repel low surface energy liquids, surface chemistry and micro/nanostructure are the two key parameters that must be optimized [75,76]. The surface chemistry of a fabric coating can improve the surface’s ability to resist wetting by low surface tension liquids. Omniphobic materials are characterized by a surface chemistry that is both hydrophobic and oleophobic, with contact angles between 90° and 150° [77]. However, there are limits of omniphobicity. For instance, C8-coated surfaces possess the lowest free energy and the best hydrophobicity of any coatings, but the maximum contact angle only reaches approximately 120° on flat surfaces [78].

On the other hand, the lotus leaf, well-known for the self-cleaning effect described earlier, exhibits a water contact angle of over 160° and a sliding angle as small as 2°. The large contact angle and small sliding angle are attributed to the wax coating on the micro/nanoscale structure on the leaf [4]. The lotus leaf exemplifies the importance of the surface topographic structure and its influence on the liquid repellence, as described by the modified equation [49,79,80]
cos*θ*′ = fr cos *θ* − (1 − f),(1)
where *θ*′ is the apparent contact angle on a rough surface, *θ* is the intrinsic contact angle on a flat surface, f is the fraction of the solid/liquid interface, and r is roughness factor, while (1 − f) is the fraction of the air/liquid interface. This equation indicates that when a topographically rough surface comes into contact with a liquid, air trapped in the rough area may occur, greatly increasing the omniphobicity [81,82]. That is, an additional layer of surface roughness is necessary to increase the fractional contact of the solid/liquid interface, and this micro/nano surface roughness can be engineered by adding micro/nanoparticles to the coating solution.

As described, the OmniBlock-coated cotton fabrics were created by combining fluoro surface chemistry and micro/nano surface roughness via sol-gel chemistry and showed excellent omni-liquid repellency. Like the lotus leaf, we have shown that the surface chemistry and micro/nanostructure can be tailored to improve the liquid repellence properties. Using our sol-gel derived fluoropolymer-coated Si NPs coating called OmniBlock, we showed that the surface of cotton can be improved significantly. OmniBlock-coated cotton fabrics clearly exhibited a much different surface chemistry and micro/nano surface roughness than raw cotton and achieved excellent omni-liquid repellency.

It should be noted that OmniBlock-coated cotton was fabricated by one-step process using PFOA-free fluoropolymer. For comparison, Yeerken et al. reported two-step process to make the superhydrophobic surface with the static contact angle of 154° and rolling off angle of 4° [83]. They used dip and spray coating of PTFE and silica particles which was followed by PTFE membrane hot pressing. Additionally, Jung et al. have studied chemical warfare agent HD-repellent cotton using UiO-66-NH_2_ or Zr(OH)_4_ and hydrophobic aminopropylisooctyl polyhedral oligomeric silsesquioxane (O-POSS) [84]. Although, the coated fabrics exhibited the static contact angle >150° for a 5-μL water droplet, 107° for a 3-μL HD droplet and roll-off angle of 7 for 50 μL. The process still requires two-step coating.

### 3.3. Mechanical Properties of OmniBlock-Coated Cotton Fabric

As shown in strength-strain curves in Figure 7, the maximum strength and strain were increased from 27.8 N and 11.4% (raw cotton) to 30.5 N and 12.7% after OmniBlock coating. It is noted that the strength was improved due to the presence of siloxane group between fibers which may increase the binding strength [85]. Additionally, it should be noted that when the OmniBlock-coated cotton fabric was twisted, it shows a great flexibility similar to the untreated raw cotton fabric (Appendix A).

Finally, OmniBlock-coated cotton fabrics were washed as described in Materials and Methods and contact angles of water and n-dodecane were analyzed. As depicted in Figure 8, the contact angles of water and n-dodecane were 150° and 90°, respectively, after washing treatment. The results show that superhydrophobicity and oleophobicity of OmniBlock-coated cotton surfaces were successfully maintained with durability, while cotton fabrics coated with trichlorododecylsilane-treated Si NPs lost its superhydrophobicity after the wash [86]. We believe that the epoxy group of GLYMO used in this study may render the coating more adhesive to cotton textiles [87].

## 4. Conclusions

We proposed a simple approach for achieving an omniphobic fabric coating that repels oil, water, and CWAs by designing PFOA-free fluoropolymer-coated Si NPs. We named this coating OmniBlock, consisting of TEOS, GLYMO, Si NPs, and a PFOA-free fluoropolymer that was produced via sol-gel chemistry. The PFOA-free fluoropolymer-coated Si NPs were formed via covalent bonds by cross linking between the Si-O-Si groups. The OmniBlock-coated cotton fabric showed outstanding omniphobic properties with a water contact angle of 154°, a shedding angle of 5°, and contact angle hysteresis of 3.2°. The contact angles for n-dodecane, HD, GD, and VX were 121°, 129°, 72°, and 87° each. In particular, coated fabrics provide an adequate omniphobic performance against n-dodecane and HD. Even after the wash, the OmniBlock coating was durable with the contact angles of 150° (water) and 90° (n-dodecane). Additionally, the coating contributed to enhance the thermal stability and mechanical strength of cotton fabric. Furthermore, this coating method can be easily applied to large scale production as demonstrated in this study, and therefore could be effectively applied to a wide range of textile products, including chemical protective clothing, shelters such as tents, as well as other commercial textiles and upholstery products.

## Figures and Tables

**Figure 1 polymers-12-01826-f001:**
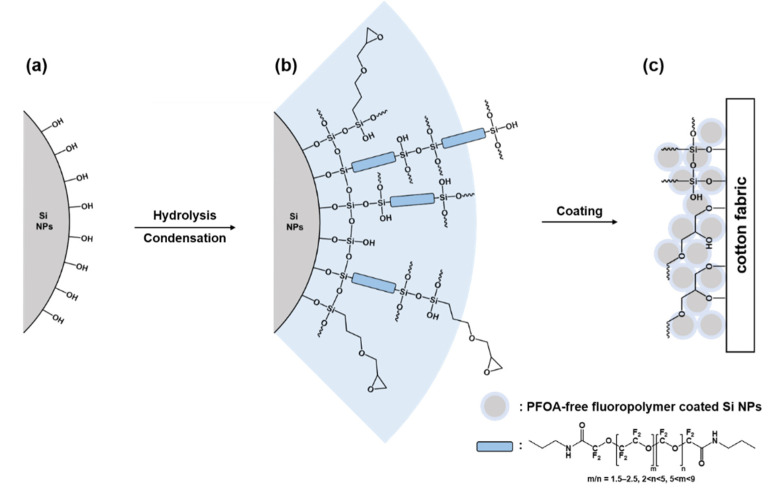
Schematic representation of OmniBlock coating onto cotton fabric (**a**) Si NPs from TEOS, (**b**) PFOA-free fluoropolymer-coated Si NPs (OmniBlock), and (**c**) dip–dry–cure results.

**Figure 2 polymers-12-01826-f002:**
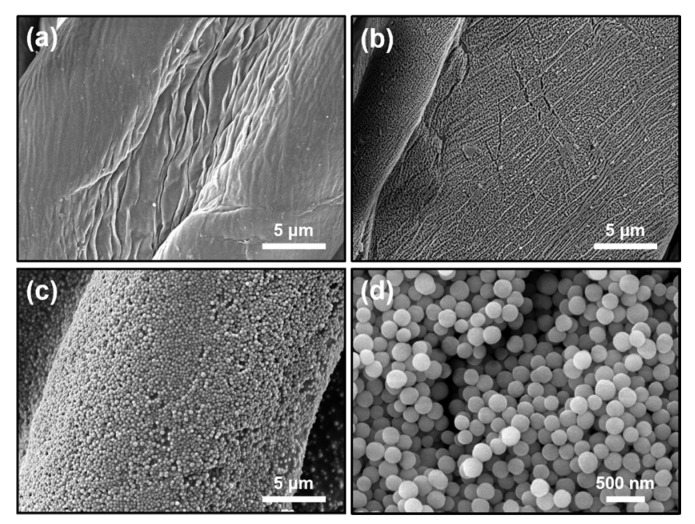
SEM images of (**a**) raw cotton, (**b**) Ar plasma-treated cotton, (**c**) OmniBlock-coated cotton fabric, and (**d**) Si NPs as prepared.

**Figure 3 polymers-12-01826-f003:**
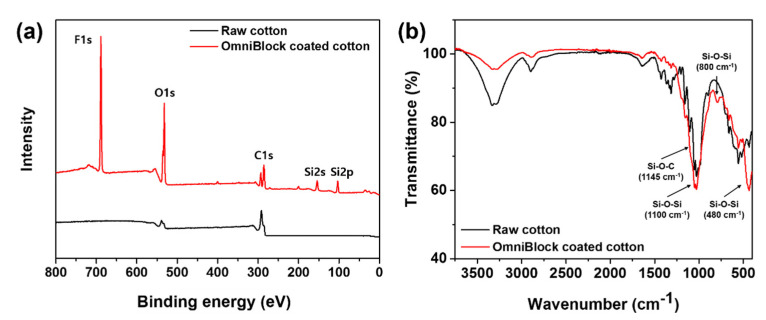
(**a**) XPS spectra of and (**b**) ATR-FTIR spectra of raw cotton fabric and OmniBlock-coated cotton fabric.

**Figure 4 polymers-12-01826-f004:**
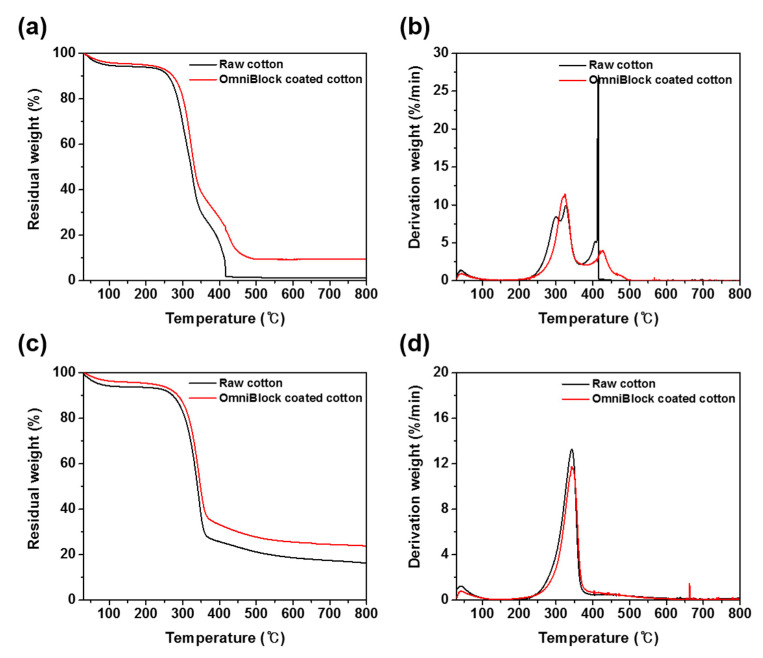
TGA and DTG curves of raw and OmniBlock-coated cotton fabrics (**a**,**b**) in air and (**c**,**d**) nitrogen.

**Figure 5 polymers-12-01826-f005:**
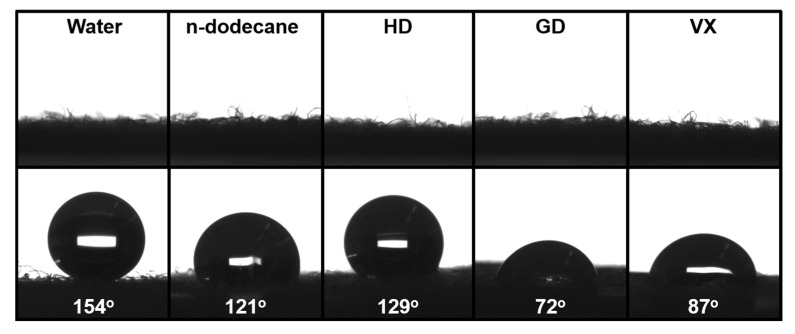
The images and contact angles of static droplets on raw cotton fabric (**Top**) and OmniBlock-coated cotton fabric (**Bottom**) from the drop shape analyzer. Water (5 μL) at *θ* = 154°, n-dodecane (5 μL) at *θ* = 121°, HD (3 μL) at *θ* = 129°, GD (3 μL) at *θ* = 72°, and VX (3 μL) at *θ* = 87°.

**Figure 6 polymers-12-01826-f006:**
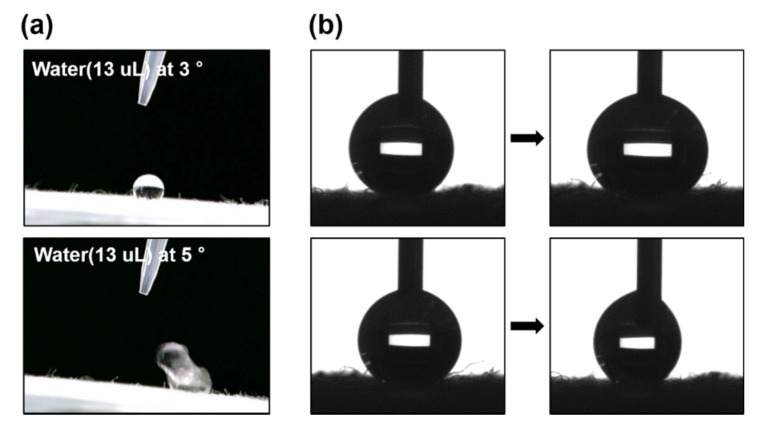
The captured images of (**a**) shedding angles for water and (**b**) the advancing angle measurement (**top**) and the receding angle measurement (**bottom**) for contact angle hysteresis.

**Figure 7 polymers-12-01826-f007:**
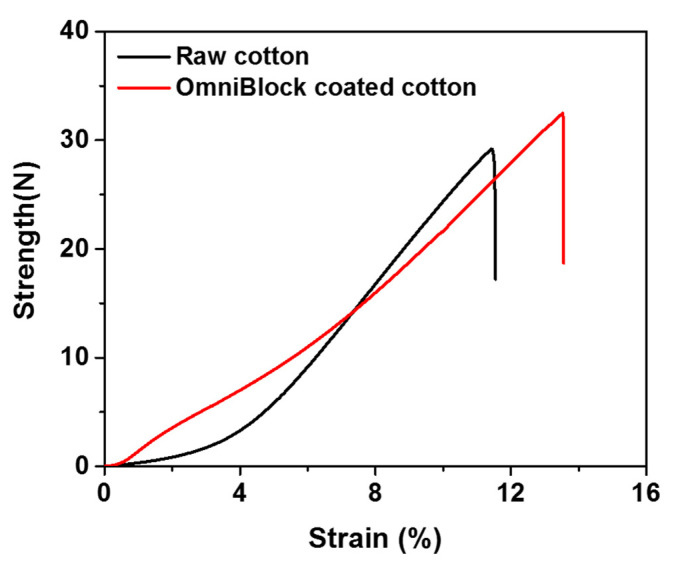
Strength–strain curves of raw cotton (in black) and OmniBlock-coated cotton (in red).

**Figure 8 polymers-12-01826-f008:**
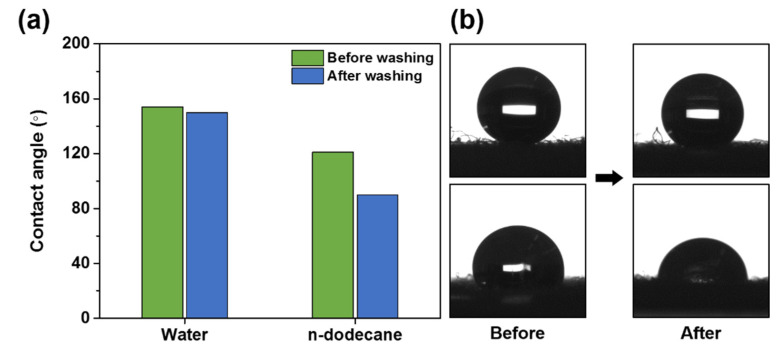
The contact angles (**a**) and captured images (**b**) of water droplet (**top**) and n-dodecane droplet (**bottom**) on OmniBlock-coated cotton fabric after washing treatment.

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
