# Peer review of "Micro/Nanostructured Coating for Cotton Textiles That Repel Oil, Water, and Chemical Warfare Agents"

_polymers, 2020, doi:10.3390/polym12081826_

Round 1
Reviewer 1 Report
Authors are recommended to improve the following before the manuscript can be published.
- Authors are well aware of the drawback of the other surface treatments for omniphobic effects, which is "the resultant surfaces are easily damaged". However, no efforts were reported in this regard.
- Raw cotton fabric specifications were missing, e.g. the construction details, unit mass, yarn counts, etc.
- How can a solution be rinsed? (line 98). "Solution" is a typo. Overall the rinse process has to be described more clearly.
- The size characterization of Si NP needs to be presented.
- The "densely packed" coated Si NP can not be judged by the appearance only. What does "densely packed" mean? Do authors have any support data?
- The TGA results do not relate to the omniphobic performance
and can be omitted thought the remaining weight is an
indication of the total Si NP add-on, which is about 5%
of the sample presented. - Any static contact angle smaller then 90 degree
means the surface is wettable by the liquid. Therefore GD and VX can wet the coated fabric, which means the coated fabric does not provide adequate omniphobic performance against GD and VX. The wording in the conclusions should be changed accordingly. - An uncoated raw cotton fabric surface, as a control, should also be tested with all liquids used to show the improvement by the presented treatment.
- Authors did not present any data related to the surface's nano structure. Therefore any wording in the discussion about the "nano" structure should be removed.
- Before the mechanical strength of the coated Si NP on the fabric can be tested to support their durability in wear and laundering, this process has a long way to go for the practical applications.
-
It is recommended to move Figures S1 and S2 to the main text of the manuscript with additional discussion to be added.
-
It is also recommended to move Figures S4 to the main text with the addition of the shedding angles of the other liquids to make the omniphobic effect achieved more convincing.
Reviewer 2 Report
The authors have presented an interesting study that will be of interest to the journal readership. It is requested that the following issues be addressed before it is accepted for publication.
The authors state that the ATR-FTIR spectra confirm covalent attachment of the treatment agent to fabric. Please elaborate on what specific features in the spectra point to a covalent attachment.
There is no discussion of the treatment durability. Please describe what is known of the wash and light fastness of the applied finish.
Reviewer 3 Report
The paper from Kwon et al. reports on the design of hydrophobic and oleophobic coatings obtained by combining silica nanoparticles (from tetraethylorthosilicate), 3-glycidoxypropyltrimethoxysilane, and a perfluorooctanoic acid-free fluoropolymer (commercially available). To this aim, a sol-gel process is exploited. The novelty of the manuscript is somehow limited, as the proposed treatments have been already exploited on cotton and polyester fabrics (there exist also some commercially-available treatments), and the conclusions are not well supported by the experimental data. Therefore, the manuscript does not seem to be suitable for publication in Polymers and should be rejected.
Some comments and suggestions are listed as follows:
- Title should contain the target of the surface treatments, i.e. cotton
- The Itroduction should better highlight the step-forward of the proposed work with respect to the already existing scientific literature. This latter is plenty of nice work on the proposed topic
- TG analyses should also be performed in air, and the obtained results commented
- Dynamic contact angle measurements should be performed instead of static measurements: in doing so, the advancing and receding contact angle values could be evaluated and the hysteresis, i.e. the difference between advancing and receding angles, could be calculated
- FTIR-ATR spectroscopy is too qualitative for assessing the formation of covalent bonds between silica nanoparticles and the fluropolymer. Conversely, solid state NMR analyses should be performed, and the degree of condensation evaluated
- Figure 4: please add the dTG (i.e. derivative curves) and comment the Tmax temperatures; besides, a Table collecting all the TG data (Tonset, Tmax, T10%, residue at Tmx, final residue) should be added to the text.
- The environmental-friendly character of any chemical product such as a fluoropolymer is very questionable
- What about the hand (i.e. soft touch) of the treated fabrics? Some tensile tests should be performed
- Durability of surface treatments on fabrics is always a key issue, as most of the fabrics have to withstand washing cycles. Therefore, the washing fastness of the fabrics should be assessed. Besides, the wettability tests should be repeated.
- From an overall point of view, the Discussion section is plenty of results but lacks of a critical interpretation/justification of the obtained data. Therefore it should be re-written completely
Round 2
Reviewer 3 Report
The manuscript has been revised according to most of the Reviewers' comments and suggestions; now it seems suitable for publication in Polymers.